# Tyrosine phosphorylation of lamin A by Src promotes disassembly of nuclear lamina in interphase

Ching-Tung Chu[1,2] , Yi-Hsuan Chen[3] , Wen-Tai Chiu[4] , Hong-Chen Chen[1,2,1]

**Lamins form the nuclear lamina, which is important for nuclear structure and activity. Although posttranslational modifications, in particular serine phosphorylation, have been shown to be important for structural properties and functions of lamins, little is known about the role of tyrosine phosphorylation in this regard. In this study, we found that the constitutively active Src Y527F mutant caused the disassembly of lamin A/C. We demonstrate that Src directly phosphorylates lamin A mainly at Tyr45 both in vitro and in intact cells. The phosphomimetic Y45D mutant was diffusely distributed in the nucleoplasm and failed to assemble into the nuclear lamina. Depletion of lamin A/C in HeLa cells induced nuclear dysmorphia and genomic instability as well as increased nuclear plasticity for cell migration, all of which were partially restored by re-expression of lamin A, but further promoted by the Y45D mutant. Together, our results reveal a novel mechanism for regulating the assembly of nuclear lamina through Src and suggest that aberrant phosphorylation of lamin A by Src may contribute to nuclear dysmorphia, genomic instability, and nuclear plasticity.**

## Introduction

Lamins (lamin A/C, B1, and B2) are type V intermediate filament proteins that form the nuclear lamina underlying the nuclear envelope (Goldman et al, 1986; McKeon et al, 1986). The nuclear lamina provides mechanical strength to the nucleus and supports various nuclear activities, including transcription, DNA replication, and DNA damage repair, which occur through interaction with chromatins and signaling proteins (Dittmer & Misteli, 2011; Ho & Lammerding, 2012). The genomes of mammals have three lamin genes: *LMNA, LMNB1,* and *LMNB2.* The *LMNA* gene is expressed in differentiated cells, whereas at least one *LMNB* gene is expressed in every somatic cells in the body (Peter et al, 1989; Lin & Worman, 1993). The *LMNA* gene produces two major isoforms through alternative splicing: lamins A and C (Fisher et al, 1986; Mckeon et al,

1986; Lin & Worman, 1993). They are identical for the first 566 amino acids, but lamin C lacks 98 amino acids and has 6 unique amino acids at the carboxyl terminus. Prelamin A (664 amino acids), the precursor of lamin A, has 98 unique carboxyl-terminal amino acids that contain a CAAX motif (Weber et al, 1989; Hennekes & Nigg, 1994). The CAAX motif of lamin A is modified by farnesylation and is important for targeting the inner nuclear membrane (Gelb et al, 2006). The functional diversification of A- and B-type lamins has been interpreted through their differences in protein structure, expression, localization patterns, and biochemical properties (Rober et al, 1989; Lammerding et al, 2006; Adam & Goldman, 2012; Nmezi et al, 2019). Mutations in the lamin genes that affect nuclear lamina assembly are associated with a group of diseases collectively referred to as laminopathies (Worman & Bonne, 2007; Kang et al, 2018).

Following the discovery that lamins are reversely disassembled during mitosis (Gerace & Blobel, 1980), early studies focused on lamin phosphorylation during the process. CDK1-mediated serine phosphorylation was found to lead to mitotic disassembly of the nuclear lamina 30 yr ago (Heald & McKeon, 1990; Peter et al, 1990). Later, serine phosphorylation of lamins was also found to be present during the interphase and important for their structural properties and functions (Kochin et al, 2014; Torvaldson et al, 2015). In addition, sumoylation was found to regulate lamin A function (Zhang & Sarge, 2008) and its interaction with retinoblastoma protein (Sharma & Kuehn, 2016). More recently, acetylation of lamin A/C was reported to be important for maintaining nuclear architecture and genome integrity (Karoutas et al, 2019).

Lamin A is a heavily phosphorylated protein with more than 70 identified unique Ser/Thr phosphorylation sites (Simon & Wilson, 2013; Machowska et al, 2015), some of which have been shown to determine the structure and function of lamin A during interphase and mitosis (Torvaldson et al, 2015). However, little is known about the role of tyrosine phosphorylation in this regards. The epidermal growth factor receptor (EGFR) was reported to phosphorylate lamin A at several tyrosine residues in vitro, including Y45, Y81, Y211, Y359, Y376, Y481, and Y646 (Tsai et al, 2015). Among which, Y45 and Y481 are known to be associated with laminopathies and predicated to be

[1]Institue of Biochemistry and Molecular Biology, National Yang Ming Chiao Tung University, Taipei, Taiwan  [2]Cancer Progression Research Center, National Yang Ming Chiao Tung University, Taipei, Taiwan  [3]Department of Life Sciences, National Chung Hsing University, Taichung, Taiwan  [4]Department of Biomedical Engineering, National Cheng Kung University, Tainan, Taiwan

Correspondence: hcchen1029@nycu.edu.tw

phosphorylation (Lin et al, 2020). Nevertheless, tyrosine phosphorylation has been reported in other intermediate filaments. For example, keratin 8, a type II intermediate filament protein, was shown to be phosphorylated on Tyr267 in the rod domain, which decreases the solubility of keratin 8 filaments (Snider et al, 2013). Although the tyrosine kinase responsible for the Tyr267 phosphorylation is uncertain, this phosphorylation site is targeted by the phosphatase PTP1B (Snider et al, 2013). In addition, keratin 19, a type I intermediate filament protein, was reported to be phosphorylated at Tyr391 in the tail domain in the presence of Src or when treated with pervanadate (Feng et al, 1999; Zhou et al, 2010), but the functional significance of this remains unknown.

Src is a non-receptor protein tyrosine kinase that has been implicated in a wide variety of cellular functions, including cell proliferation, survival, and migration (Martin, 2001; Yeatman, 2004). Its increased expression or activity has been associated with the malignant progression of many human tumors (Irby & Yeatman, 2000; Summy & Gallick, 2003). Src mainly exerts its functions at the plasma membrane and cytoplasm but can be detected in the nucleus (Bagnato et al, 2020). However, the nuclear function of Src remains obscure. Only a few nuclear proteins are known to interact with Src. For example, Src interacts with the transcription factor heterogeneous nuclear ribonucleoprotein K (hnRNPK) and RNA-binding protein Sam68, which may influence the transcription and processing of pre-mRNAs (Gondran & Dautry, 1999; Hartmann et al, 1999; Shen et al, 1999). In addition, Src was found to phosphorylate nuclear membrane proteins emerin and LAP2-β (Tifft et al, 2009), both of which are lamin-associated proteins. The tyrosine phosphorylation of emerin is important for its binding to barrier-to-autointegration factor (BAF), a conserved chromatin protein that is essential for cell division (Tifft et al, 2009).

More recently, we found that Src phosphorylates vimentin, a type III intermediate filament protein, and regulates the dynamics and organization of vimentin filaments during cell migration (Yang et al, 2019). The Tyr117 of vimentin was identified to be the major phosphorylation site for Src (Yang et al, 2019). The Tyr117 and its flanking sequences are conserved in some types of intermediate filament proteins such as lamins, which raises the intriguing possibility that Src may phosphorylate lamins and regulate their assembly. In this study, we set out to explore this possibility and demonstrate that Src phosphorylates lamin A mainly at Tyr45, which has an adverse effect on lamin A assembly. More importantly, our results suggest that aberrant phosphorylation of lamin A by Src may contribute to nuclear dysmorphia, genomic instability, and nuclear plasticity.

# Results

## Constitutively active Src causes disassembly of lamin A/C in intact cells

To examine the potential effect of Src on the nuclear lamina, the constitutively active Src Y527F mutant (Src Y527F) was transiently expressed in MCF7 human breast cancer cells (Fig 1), which retains several characteristics of differentiated mammary epithelium (Comsa

et al, 2015). We found that Src Y527F induced tyrosine phosphorylation and disassembly of lamin A/C in MCF7 cells (Figs 1A and B and S1). Similar effects were also observed in Madin-Darby canine kidney epithelial cells (Fig S2A and B). Compared with lamin A/C, lamin B1 was less affected by Src Y527F in MCF7 cells (Fig 1C). Unlike Src Y527F that caused breakdown and aggregation of lamin A/C, HA epitope–tagged c-Src (HA-cSrc) mainly caused a partial dispersion of lamin A/C (Fig 1D). However, the sequestration of HA-cSrc in the nucleus by fusion with a NLS apparently induced a more severe consequence, leading to breakdown and aggregation of lamin A/C (Fig 1D). The targeting of HA-cSrc to the endoplasmic reticulum and Golgi apparatus had no such effect (Fig S3). Together, these results suggest that Src may phosphorylate lamin A in the nucleus, which leads to disassembly of lamin A/C. However, the possibility that the aggregation of lamin A/C may arise during post-mitotic nuclear re-assembly cannot be excluded.

## Src directly phosphorylates lamin A at Tyr45

The tyrosine phosphorylation of lamins A, B1, and B2 was examined in HeLa cells. Among which, lamin A was readily detected to be tyrosine phosphorylated (Fig 2A). To immunoprecipitate both lamin A and lamin C, an anti-lamin A/C antibody was generated. With this antibody, we demonstrated that lamin C was barely tyrosine phosphorylated in HeLa cells (Fig 2B). To examine whether Src phosphorylates lamin A, green fluorescent protein–fused lamin A (GFP-lamin A) was transiently co-expressed with Src Y527F in HEK293 cells. Indeed, GFP-lamin A were highly tyrosine phosphorylated by Src Y527F (Fig 2C), and the Tyr45 residue of lamin A was identified to be phosphorylated by mass spectrometry (Fig 2D). We also demonstrated that Src directly phosphorylated purified His-tagged lamin A (His-lamin A) in vitro (Fig 2E). The substitution of lamin A Tyr45 with Phe caused ~40–50% decrease in the tyrosine phosphorylation of lamin A induced by Src both in vitro (Fig 2E) and in intact cells (Fig 2F).

To facilitate the detection of Tyr45-phosphorylated lamin A, a phospho-specific antibody (anti-lamin pY45) was generated, which recognized Src-phosphorylated lamin A, but not the Y45F mutant (Fig 2G). The specificity of this antibody to Tyr45-phosphorylated lamin A was demonstrated by successful blocking with a phosphopeptide with the sequence derived from lamin A Tyr45 (Fig 2G). With this antibody, we showed that Src Y527F induced Tyr45 phosphorylation of endogenous lamin A in $LMNA^{+/+}$ HeLa cells, but not in $LMNA^{-/-}$ HeLa cells (Figs 2H and S4). In addition, the inhibition of tyrosine phosphatases by sodium orthovanadate ($Na_3VO_4$) increased Tyr45 phosphorylation of endogenous lamin A in HeLa cells (Fig S5A) and MDA-MB-231 cells (Fig S5B).

## Phosphorylation of lamin A at Tyr45 may prevent its assembly into the nuclear lamina

The Tyr45 resides at the coil 1A subdomain of lamin A. To examine the effect of Tyr45 phosphorylation on the polymerization of lamin A, Tyr45 was substituted with Asp or Phe. His-lamin A and the mutants (Y45D and Y45F) were expressed in and purified from *Escherichia coli* (Fig 3A). As reported previously (Peter et al, 1991), His-lamin A was soluble in buffer with 500 mM of sodium chloride but became insoluble upon reduction of the concentration of

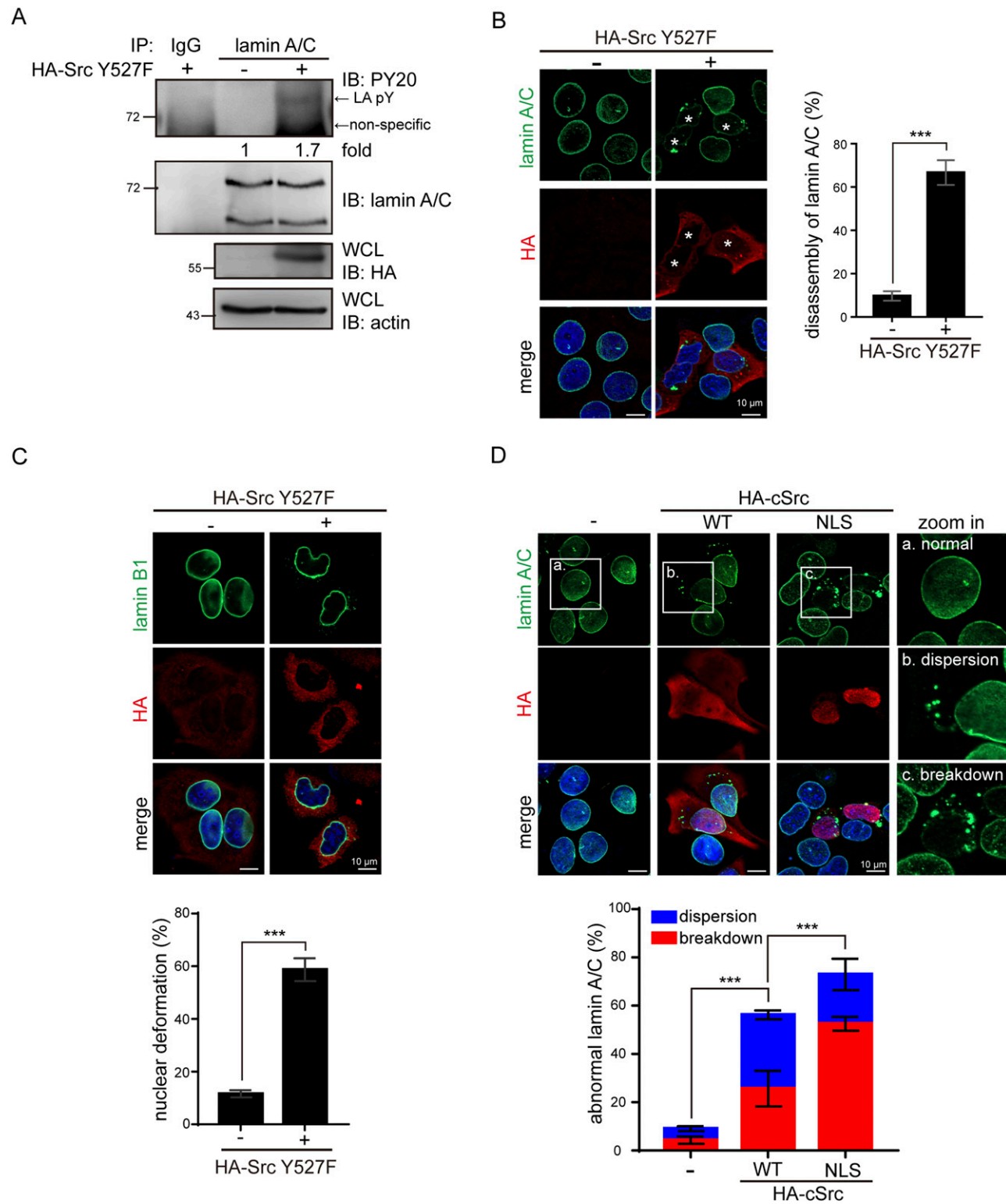

**Figure 1. Constitutively active Src causes disassembly of lamin A/C.**
**(A)** MCF7 cells were transiently transfected with (+) or without (−) HA-Src Y527F. The endogenous lamins A and C was immunoprecipitated (IP) with anti-lamin A/C or pre-immune serum (IgG) as a control. The immunonocomplexes were analyzed by immunoblotting (IB) with anti-phosphotyrosine (PY) or anti-lamin A/C. An equal amount of whole cell lysates was analyzed by immunoblotting with anti-HA or anti-actin. Tyrosine phosphorylated lamin A (LA pY) was indicated by arrow. The tyrosine phosphorylation of lamin A was measured and expressed as −fold relative to the level without Src Y527F. **(B)** MCF7 cells were transiently transfected with (+) or without (−) HA-Src Y527F. The cells were fixed and stained for lamin A/C, HA-Src, and DNA. The asterisks indicate the cells with Src Y527F expression. Scale bars, 10 $\mu$m. The percentage of Src Y527F-transfected cells with disassembly of lamin A/C was measured (n ≥ 300). Values (means ± SD) are from three experiments. ***$P$ < 0.001. **(C)** MCF7

sodium chloride to 150 mM (Fig 3B). Interestingly, a large fraction of the phosphomimetic Y45D mutant remained soluble even in the presence of 150 mM sodium chloride (Fig 3B). To examine the assembly of FLAG-tagged lamin A and the mutants in intact cells, the endogenous lamin A/C in HeLa cells was depleted by the CRISPR/Cas9 system (Fig 3C). The depletion of lamin A/C caused an elongated nuclear shape and the loss of nuclear envelope integrity, as manifested by abnormal cytoplasmic distribution of emerin (Fig 3D). Transient expression of FLAG-lamin A in LMNA$^{-/-}$ HeLa cells resulted in assembly into nuclear lamina (Fig 3E) and the restoration of the nuclear shape (Fig 3H). However, the Y45D mutant failed to assemble into the nuclear lamina. Instead, it diffusely distributed in the nucleoplasm (Fig 3E–G) and further increased the ratio of elongated nuclei in LMNA$^{-/-}$ HeLa cells (Fig 3H). These data support that phosphorylation of lamin A at Tyr45 may have an adverse effect on its assembly.

The phosphorylation of lamin A/C at Ser22 is already known to have an adverse effect on lamin A/C assembly during mitosis and interphase (Heald & McKeon, 1990; Peter et al, 1990; Kochin et al, 2014). In this study, we found that the S22D mutant had better capabilities to assemble into nuclear lamina and restore the nuclear shape of LMNA$^{-/-}$ HeLa cells than the Y45D (Fig 3E–H). These results suggest that the phosphorylation at Tyr45 may have a more profound effect on the lamin A assembly than the phosphorylation at Ser22. Next, the possibility of a reciprocal regulation between the Tyr45 and Ser22 phosphorylation was examined. We found that the substitution of the Tyr45 with Asp or Phe did not affect the phosphorylation of lamin A at Ser22 during interphase and mitosis (Fig 3I). Importantly, we found that the Tyr45 phosphorylation of lamin A was apparently increased in mitosis, accompanied by increased Src activity (Fig 3J). The mitosis-increased Tyr45 phosphorylation of lamin A was substantially lower in the S22A mutant (Fig 3J), suggesting that prior phosphorylation at Ser22 may be required for Tyr45 phosphorylation during mitosis.

### Phosphorylation of lamin A at Tyr45 may be crucial for its dynamics

To examine the dynamics of GFP-lamin A and its Tyr45 mutants in live cells, two fluorescence microscopy techniques—fluorescence loss in photobleaching (FLIP) and FRAP—were performed in HeLa cells. The Y45D mutant was diffusible and very dynamic, exhibited much faster FLIP and FRAP than the WT and Y45F mutant. Within 4 min, the Y45D mutant lose more than 80% by FLIP (Fig 4A and B) and recovered ~80% by FRAP (Fig 4C and D). The Y45F mutant had FLIP and FRAP similar to the WT (Fig 4A–D). Notably, the WT, but not the Y45F mutant, had ~5% of FRAP within 4 min after photobleaching (Fig 4E). These results together suggest that the phosphorylation of lamin A at Tyr45 may regulate its dynamics.

### Aberrant phosphorylation of lamin A by Src may cause nuclear dysmorphia

To examine the effect of Src-mediated phosphorylation of lamin A on nuclear morphology, human breast cancer cell lines MCF7 and MDA-MB-231 were treated with the selective Src inhibitor dasatinib. The MCF7 cell line retains several characteristics of differentiated mammary epithelium (Comsa et al, 2015), whereas the MDA-MB-231 cell line is a highly aggressive, invasive, and poorly differentiated line of triple-negative breast cancer cells (Chavez et al, 2010). Compared to MCF7 cells, MDA-MB-231 cells had higher Src activity (Fig 5A) accompanied by higher lamin A tyrosine phosphorylation (Fig 5A). MDA-MB-231 cells displayed nuclear dysmorphia with apparent nuclear lobulation (Fig 5B). Inhibition of the Src activity by dasatinib decreased the tyrosine phosphorylation of lamin A (Fig 5A) and restored the nuclear circularity and shape in MDA-MB-231 cells (Fig 5B). Moreover, overexpression of FLAG-lamin A and the Y45F mutant in MDA-MB-231 cells partially restored the nuclear shape (Fig 5C and D). In contrast, the Y45D mutant increased the extent of nuclear dysmorphia (Fig 5C and D). These results suggest that aberrant phosphorylation of lamin A by Src may lead to nuclear dysmorphia.

### Aberrant phosphorylation of lamin A by Src may cause genomic instability

To visualize the effect of the lamin A Tyr45 mutants on nuclear morphology in live cells, mCherry-lamin A and the Y45 mutants were transiently co-expressed with GFP-H2B in LMNA$^{-/-}$ HeLa cells and monitored with time-lapse microscopy (Fig 6A). The mCherry alone did not affect the nuclear shape of LMNA$^{+/+}$ HeLa cells (Videos 1). Approximately 35% of LMNA$^{-/-}$ HeLa cells displayed "unstable" nuclei, which were characterized by constant changes in their nuclear shape with concaves or lobules (Videos 2). The reexpression of mCherry-lamin A and the Y45F mutant into LMNA$^{-/-}$ HeLa cells partially restored their nuclear stability and shape (Videos 3 and 4). In contrast, the Y45D mutant further increased the ratio of the "unstable" nuclei (Fig 6A and Videos 5) and eventually caused cell death in ~25% of the cells (Fig 6A and Videos 6).

Micronucleus, one of the characteristics of genomic instability (Kalsbeek & Golsteyn, 2017), was detected in ~7% of LMNA$^{-/-}$ HeLa cells (Fig 6B). The re-expression of FLAG-lamin A partially rescued the defect. However, the Y45D mutant deteriorated the genomic stability, leading to ~20% of the cells having micronuclei (Fig 6B). To examine whether lamin A/C depletion affects DNA repair capability, mitomycin C was used to induce DNA double-strand breaks (Mladenov et al, 2007). The DNA repair response was measured by γ-H2AX foci (Sedelnikova et al, 2002). The quantitation of γH2AX foci has been used as a marker of DNA damage and repair in the context of DNA double-strand breaks (Paull et al, 2000). Our results showed

---

cells were transiently transfected with (+) or without (−) HA-Src Y527F. The cells were fixed and stained for lamin B1, HA-Src, and DNA. Scale bars, 10 μm. The percentage of Src Y527F-transfected cells with nuclear deformation was measured (n ≥ 300). Values (means ± SD) are from three experiments. ***P < 0.001. **(D)** MCF7 cells were transiently transfected with HA-cSrc, HA-cSrc-NLS, or vector alone. The cells were fixed and stained for lamin A/C, HA-cSrc, and DNA. Representative images for the cells with normal (a), dispersion (b), or breakdown (c) of lamin A/C are shown. Scale bars, 10 μm. The percentage of HA-cSrc transfected cells with dispersion or breakdown of lamin A/C was measured (n ≥ 300). Values (means ± SD) are from three experiments. ***P < 0.001.
Source data are available for this figure.

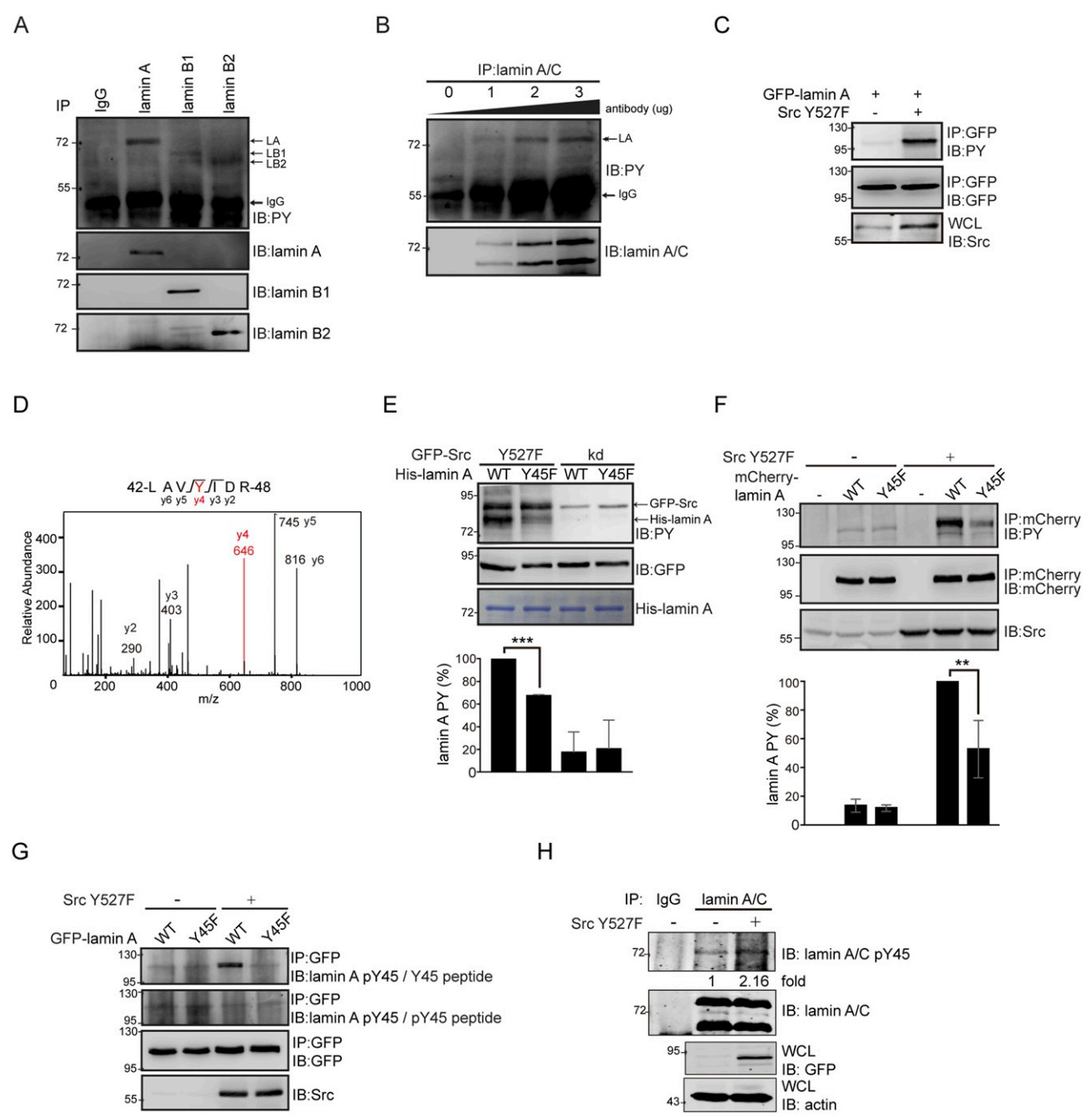

**Figure 2. Src directly phosphorylates lamin A at Tyr45.**
**(A)** HeLa cells were lysed by sonication in RIPA buffer. Lamins A, B1, and B2 were immunoprecipitated (IP) with specific antibodies or pre-immune serum (IgG) as the control. The immunonocomplexes were analyzed by immunoblotting (IB) with antibodies as indicated. **(B)** Lamins A and C in the HeLa cell lysates were immunoprecipitated (IP) with 0, 1, 2, or 3 μg of anti-lamin A/C. The immunonocomplexes were analyzed by immunoblotting (IB) with anti-PY or anti-lamin A/C. **(C)** GFP-lamin A was transiently co-expressed with (+) or without (−) Src Y527F in HEK293 cells. The expression of Src Y527F was analyzed by immunoblotting (IB) with anti-Src. GFP-lamin A was immunoprecipitated (IP) with anti-GFP and the immunonocomplexes were analyzed by immunoblotting (IB) with anti-PY or anti-GFP. An equal amount of whole cell lysates was analyzed by immunoblotting (IB) with anti-Src. **(D)** GFP-lamin A was transiently co-expressed with Src Y527F in HEK293 cells. GFP-lamin A was purified and analyzed by mass spectrometry. The result reveals lamin A Tyr45 is phosphorylated. **(E)** GFP-Src Y527F and the kinase-defective (kd) mutants were transiently expressed in HEK293 cells and their expression was analyzed by immunoblotting (IB) with anti-GFP. GFP-Src was immunoprecipitated (IP) with anti-GFP and the immunocomplexes were subjected to an in vitro kinase assay using purified His-tagged lamin A WT and Y45F mutant as the substrate. The inputs of His-lamin A WT and Y45F mutant were visualized with Coomassie Blue stain. The tyrosine phosphorylation of His-lamin A WT and Y45F was analyzed by immunoblotting (IB) with anti-PY. The tyrosine phosphorylation of His-lamin A was measured and expressed as percentage relative to His-lamin A WT with GFP-Src Y527F. Values (means ± SD) are from two experiments. ***P < 0.001. **(F)** mCherry-lamin A and the Y45F mutant were transiently co-expressed with (+) or without (−) Src Y527F in HeLa cells. mCherry-lamin A was immunoprecipitated (IP) with anti-mCherry and the immunonocomplexes were analyzed by immunoblotting (IB) with anti-PY or anti-mCherry. The expression of Src Y527F was analyzed by immunoblotting (IB) with anti-Src. The tyrosine phosphorylation of mCherry-lamin A was measured and expressed as percentage relative to mCherry-lamin A WT with Src Y527F. Values (means ± SD) are from three experiments. **P < 0.01. **(G)** GFP-lamin A and the Y45F mutant were transiently co-expressed with (+)

that the DNA repair response of LMNA$^{-/-}$ HeLa cells was slower than that of the control HeLa cells, which was partially restored by the re-expression of FLAG-lamin A and the Y45F mutant, but not the Y45D mutant (Fig 6C). These data together suggest that aberrant phosphorylation of lamin A at Tyr45 may cause genomic instability.

### Aberrant phosphorylation of lamin A at Tyr45 by Src may increase nuclear plasticity for cell migration

To examine whether the phosphorylation of lamin A Tyr45 may regulate nuclear plasticity for cell migration, FLAG-lamin A and mutants were stably re-expressed in LMNA$^{-/-}$ HeLa cells (Fig 7A) and the cells were subjected to the trans-well cell migration assay. The nuclear dysmorphia of LMNA$^{-/-}$ HeLa cells was restored by FLAG-lamin A but deteriorated by the Y45D mutant (Fig 7B). The capability of LMNA$^{+/+}$ and LMNA$^{-/-}$ HeLa cells to migrate through the membrane with 8-$\mu$m pores were similar (Fig S6). However, LMNA$^{-/-}$ HeLa cells had better capability to migrate through 5-$\mu$m pores than LMNA$^{+/+}$ HeLa cells, which was suppressed by re-expression of FLAG-lamin A and Y45F mutant, but further promoted by the Y45D mutant (Fig 7C). The reason for the cells that failed to migrate through the pores was likely because their nuclei were stuck in the pores (Fig 7C). These data suggest that aberrant phosphorylation of lamin A at Tyr45 may increase the nuclear plasticity of the cell to facilitate cell migration in a three-dimensional environment.

## Discussion

In the present study, we demonstrated lamin A is a novel substrate of Src and identified lamin A Tyr45 as the major phosphorylation site. The Tyr45 and its flanking sequences (LNDRLAV**Y$^{45}$**IDRVRSL) are well conserved among species (Table S1) and in most of the intermediate filament proteins (Table S2). Vimentin Tyr117 (LNDRFAN**Y$^{117}$**IDKVRFL), which is equivalent to lamin A Tyr45, has been shown to be the major phosphorylation site for Src, and this phosphorylation appears to prevent the assembly of vimentin into filaments and is important for growth factor-induced cell migration (Yang et al, 2019). Likewise, we demonstrated in this study that the phosphorylation of lamin A at Tyr45 by Src has an adverse effect on its assembly. In addition, our results also implicate a potential role for Src in regulating the nuclear deformability. Cellular deformability, including both cytoplasmic and nuclear deformability, is necessary for cell migration in a three-dimensional environment (Wolf et al, 2013; Yamada & Sixt, 2019). Intermediate filaments are flexible cytoskeletal structures with high tensile strength. Identification of vimentin and lamin A as the substrates of Src enhances our understanding of how cellular deformability is regulated in response to extracellular cues. In particular, Src is often activated

upon stimulation by various growth factors and cytokines. Thus, it is possible that Src may regulate nuclear rigidity or deformability by phosphorylating lamin A and other lamin-binding proteins, such as emerin (Tifft et al, 2009).

Lamin A Tyr45 and its flanking sequences (LNDRLAV**Y$^{45}$**IDRVRSL) are conserved in lamin B1 (LNDRLAV**Y$^{46}$**IDKVRSL) and lamin B2 (LNDRLAH**Y$^{40}$**IDRVRAL) as well. However, Src Y527F apparently caused the disassembly of lamin A and to a lesser extent lamin B1 (Fig 1B and C). Compared with lamins B1 and B2, lamin A was readily detected to be tyrosine phosphorylated in HeLa cells (Fig 2A). The reason for this is unclear, but it may be the differences in their posttranslational processing and localization patterns. Mature lamin A is not farnesylated at its carboxyl terminus, whereas mature lamins B1 and B2 remain farnesylated (Adam & Goldman, 2012). The carboxyl-terminal farnesyl group of lamin B1 makes it to be more adjacent to the inner nuclear membrane than lamin A/C (Nmezi et al, 2019). Moreover, lamin C was barely tyrosine phosphorylated in HeLa cells (Fig 2B), suggesting that the regulation of lamin A tyrosine phosphorylation may be different from lamin C. This supports the notion that lamin C forms separate filaments and is functionally distinct from lamin A.

The Tyr45 motif (LAVYIDR) of lamin A has 42.9% resemblance to the Src consensus phosphorylation motif (E-X-I/L/V-Y-G-E/I-F/I or E-E-I/V-Y-G-E-X-F). In addition to Src, EGFR and the insulin receptor are predicted to be able to phosphorylate lamin A Tyr45 by the NetPhos 3.1 Server. In addition, the Tyr45 and Tyr481 of lamin A were predicated to be phosphorylation (UniProt), both of which are associated with laminopathies (Lin et al, 2020). Tsai et al (2015) described a large-scale determination of absolute phosphorylation stoichiometries by motif-targeting quantitative proteomics. In their raw datasets deposited in the ProteomeXchange Consortium, we noticed that EGFR phosphorylates lamins A, B1, and B2 at several tyrosine residues, including lamin A Tyr45 and its equivalent sites at lamins B1 and B2. However, the functional consequence of lamin tyrosine phosphorylation by EGFR remains to be investigated. Previous studies have already shown that serine phosphorylation (Kochin et al, 2014; Torvaldson et al, 2015), sumoylation (Zhang & Sarge, 2008), and acetylation (Karoutas et al, 2019) of lamins are present during interphase and important for their structural properties and functions. Therefore, the regulation of nuclear architecture through posttranslational modifications of lamins could be much more sophisticated than what we know and awaits further studies.

Early studies regarding the nuclear lamina focused more on its reversible disassembly during mitosis. Until now, CDK1-mediated serine phosphorylation has been the only well-defined functional consequence of lamin's posttranslational modification in mitosis (Heald & McKeon, 1990). Src activity has been reported to increase in mitosis (Chackalaparampil & Shalloway, 1988; Park & Cartwright, 1995). In this study, we surprisingly found that Tyr45 phosphorylation of lamin A is apparently increased in mitosis, accompanied by increased Src activity (Fig 3J). Therefore, it is possible that Src-mediated

---

or without (−) Src Y527F in HeLa cells. GFP-lamin A was immunoprecipitated (IP) by anti-GFP and the immunocomplexes were analyzed by immunoblotting (IB) with anti-lamin A pY45 in the presence of the Y45 phosphopeptide (pY45 peptide) or the Y45 peptide as the control. **(H)** Lamin A/C in the HeLa cells transiently expressing with (+) or without (−) Src Y527F was immunoprecipitated (IP) with anti-lamin A/C or pre-immune serum (IgG) as the control. The immunonocomplexes were analyzed by immunoblotting (IB) with anti-lamin A pY45 or anti-lamin A/C. An equal amount of whole cell lysates was analyzed by immunoblotting (IB) with anti-GFP or anti-actin. The Y45 phosphorylation of lamin A was quantified and expressed as ratio relative to the level without Src Y527F. Source data are available for this figure.

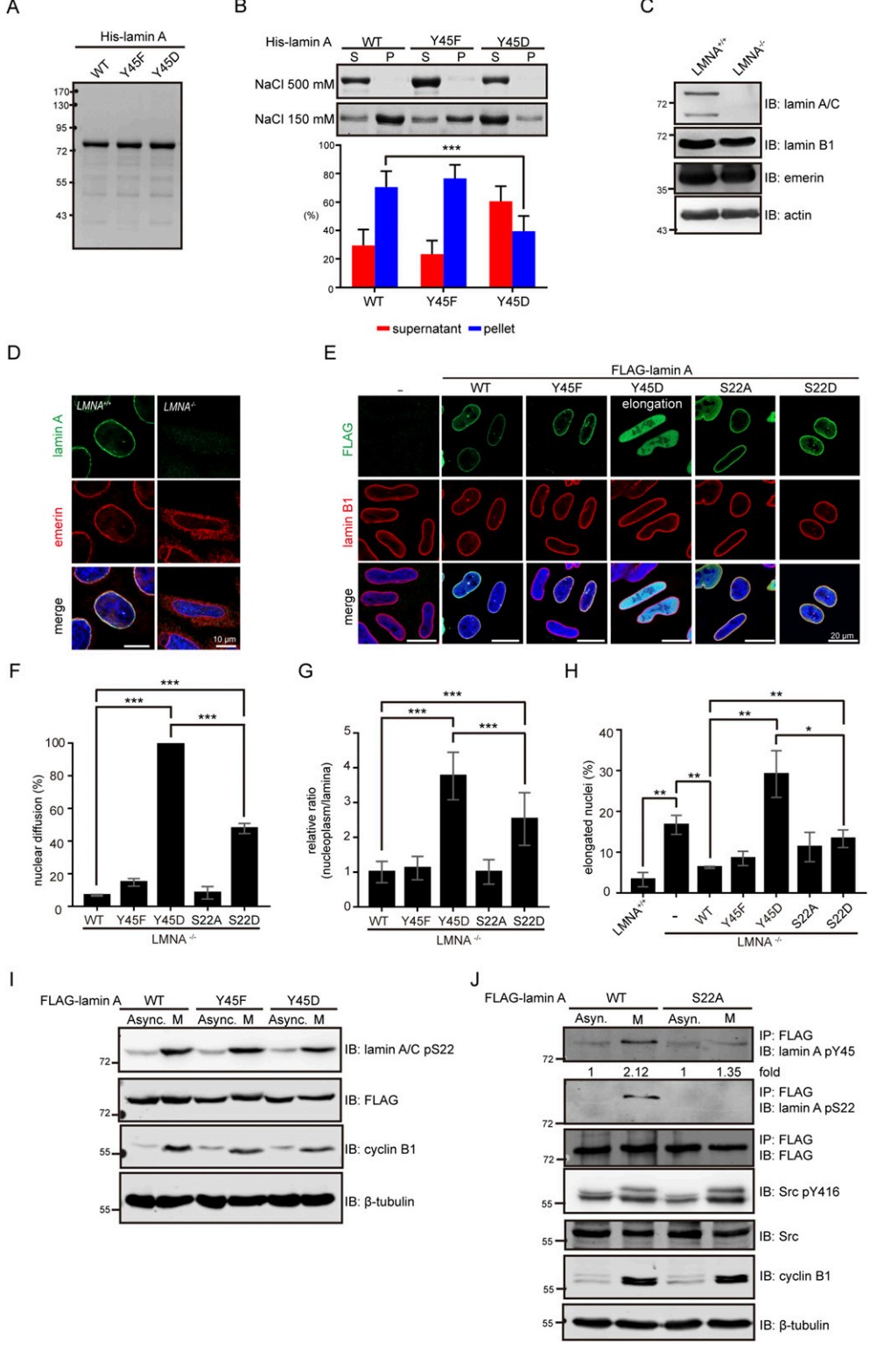

**Figure 3. Phosphorylation of lamin A at Tyr45 may prevent the assembly of lamin A.**
**(A)** Purified His-lamin A proteins were fractionated by SDS–PAGE and visualized with Coomassie Blue stain. **(B)** Purified His-lamin A proteins (0.5 mg/ml) stored in the storage buffer with 500 mM NaCl were allowed to polymerize by dialysis in the polymerization buffer with 150 mM NaCl for 3 h at room temperature. His-lamin A proteins were separated into the supernatant (S) and pellet (P) fractions by centrifugation at 14,000$g$ for 30 min. An equal proportion of His-lamin A proteins were fractionated by SDS–PAGE and visualized with Coomassie Blue stain. The ratio of lamin A polymerization was measured by Image J. Values (means ± SD) are from five independent experiments. **(C)** Lamin A/C-knockout (LMNA$^{-/-}$) HeLa cells were generated by the CRISPR/Cas9 system. An equal amount of the whole cell lysates (WCLs) from LMNA$^{+/+}$ and LMNA$^{-/-}$ HeLa cells were analyzed by immunoblotting (IB) with antibodies as indicated. **(D)** LMNA$^{+/+}$ and LMNA$^{-/-}$ HeLa cells were fixed and stained for lamin A, emerin, and DNA. Representative images are shown. Scale bars, 10 $\mu$m. **(E)** FLAG-lamin A and its mutants were transiently expressed in LMNA$^{-/-}$ HeLa cells. The cells were fixed and then stained for FLAG, lamin B1, and DNA. Representative images are shown. Scale bars, 20 $\mu$m. **(F)** The percentage of FLAG-lamin A transfected cells as described in panel E with nuclear diffusion of FLAG-lamin A was measured (n ≥ 500). ***$P < 0.001$. **(E, G)** The fluorescence intensities of FLAG-lamin A in the nucleoplasm and nuclear lamina of the cells as described in panel (E) were measured using ZEISS ZEN2 software. The relative ratios of the signals in the nucleoplasm to the signals in the lamina were calculated. Data are expressed as a ratio relative to FLAG-lamin A WT that set as one. Values (means ± SD) are from at least 50 cells. ***$P < 0.001$. **(E, H)** The percentage of FLAG-lamin A transfected cells as described in panel (E) with an elongated nucleus was measured (n ≥ 500). The elongated nucleus is defined as the long nuclear axis is threefold longer than short nuclear axis. Values (means ± SD) are from three experiments. *$P < 0.05$, **$P < 0.01$. **(I)** LMNA$^{-/-}$ HeLa cells stably expressing FLAG-lamin A or its mutants remained asynchronized (Async.) or were synchronized at the mitosis (M) by treating 200 ng/ml nocodazole for 16 h. An equal amount of WCLs was analyzed by immunoblotting (IB) with antibodies as indicated. **(J)** LMNA$^{-/-}$ HeLa cells transient expressing FLAG-lamin A or the S22A mutant remained asynchronized (Async.) or were synchronized at the mitosis (M) by treating 200 ng/ml nocodazole for 16 h. FLAG-lamin A was immunoprecipitated with anti-FLAG and the immunonocomplexes were analyzed by immunoblotting with anti-lamin A pY45, anti-lamin A pY45, or anti-FLAG. An equal amount of WCLs was analyzed by immunoblotting (IB) with antibodies as indicated. The Y45 phosphorylation of FLAG-lamin A was quantified and expressed as –fold relative to the level of FLAG-lamin A WT asynchronized. Source data are available for this figure.

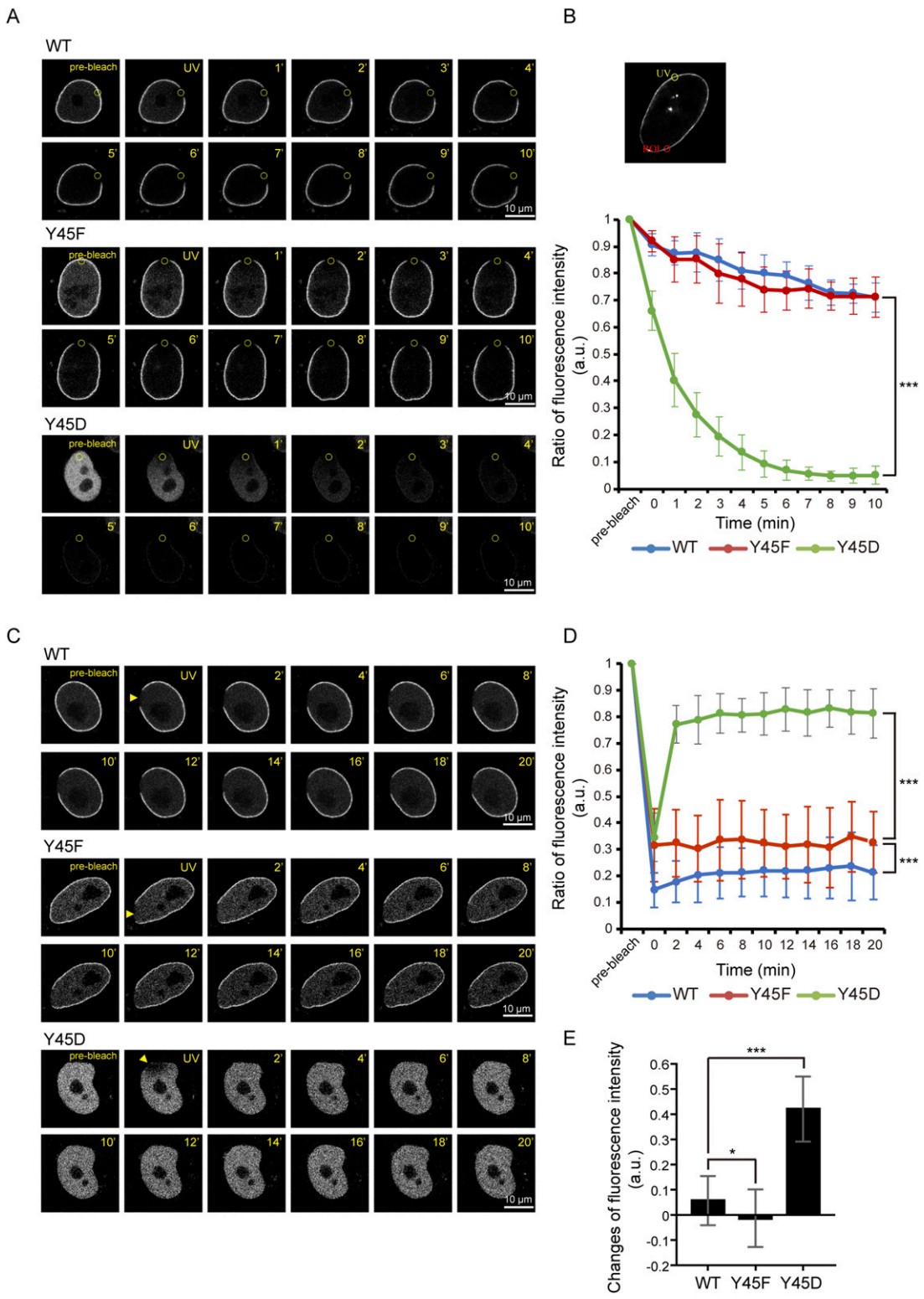

**Figure 4. The phosphorylation of lamin A at Tyr45 may affect its dynamics.**
**(A)** A fluorescence loss in photobleaching analysis was performed in HeLa cells transiently expressing GFP-lamin A WT, Y45F, or Y45D. The selected regions (indicated by yellow circle, 2-μm diameter) were photobleached by a 405-nm laser for 10 min. Representative images from confocal fluorescence microscopy (488 nm excitation) before bleaching (pre-bleach) and at different time points after bleaching are shown. Scale bars, 10 μm. **(B)** Fluorescence at the regions of interest (ROI, indicated by red circles, 2-μm diameter) was measured, and fluorescence loss in photobleaching was calculated as a ratio to the initial fluorescence. Two-way ANOVAs with Tukey's post hoc tests were used for the comparisons of WT versus Y45D at 10 min time point. The *P*-value was calculated from at least 10 cells pooled from three independent experiments.

phosphorylation of lamin A at Tyr45 may be involved in the mitotic disassembly of the nuclear lamina. The non-α-helical NH2-terminal head domain of lamin A contains several known Ser/Thr phosphorylation sites. The substitution of lamin A Tyr45 with Asp or Phe did not affect Ser22 phosphorylation during interphase and mitosis (Fig 3I), suggesting that Tyr45 phosphorylation may be not required for Ser22 phosphorylation in the head domain. However, the mitosis-increased Tyr45 phosphorylation of lamin A was substantially lower in the S22A mutant (Fig 3J), suggesting that prior phosphorylation at Ser22 may be required for Tyr45 phosphorylation during mitosis. It is possible that the serine phosphorylation in the head domain may unwind the coiled-coil structure, which thereby allowing Src to access the Tyr45 for phosphorylation during mitosis. The significance of tyrosine phosphorylation and other posttranslational modifications of lamins during mitosis required further studies.

Lamin A Tyr45 and vimentin Tyr117 reside in the coil 1A subdomain. In the case of vimentin, the coiled-coil dimers form antiparallel, half-staggered tetramers, which then assemble laterally into ~65-nm long unit-length filaments (ULFs) (Mücke et al, 2004). ULF that typically consists of eight tetramers binds end-to-end to form non-polar filaments. The Y117L variant of vimentin was shown to form ULF particles, but did not anneal longitudinally and thus could not form vimentin filaments (Meier et al, 1999). In addition, a kinase or an ATP-dependent chaperone is required for the maintence of vimentin at the ULF level in equilibrium with soluble vimentin tetramers (Robert et al, 2015). The Y117D variant of vimentin formed ULF-like particles (Yang et al, 2019), suggesting that the phosphorylation of vimentin at Tyr117 may induce the dissociation of ULF tetramers and/or prevent them from annealing longitudinally (Yang et al, 2019). It is not clear whether lamins form ULF-like structures during their polymerization. However, the crystal structure of lamin A reveals that Tyr45 residue makes an awkward interaction with Ile46 residue in another protomer (Ahn et al, 2019). Therefore, it is possible that Tyr45 phosphorylation or mutation (Y45C in Emery-Dreifuss muscular dystrophy) of lamin A may disrupt the awkward interaction and thereby prevent tetramer formation.

We showed in this study that the phosphomimetic Y45D mutant failed to assemble into the nuclear lamina and diffusely distributed in the nucleoplasm (Fig 3E). More importantly, the Y45D mutant behaved like a dominant-negative mutant which deteriorated nuclear dysmorphia, micronuclei, and impairment of the DNA repair response (Figs 5D and 6). These results suggest that aberrant phosphorylation of lamin A at Tyr45 by oncogenic Src or other tyrosine kinases may cause abnormal breakdown of nuclear lamina, which in turn leads to nuclear dysmorphia, genomic instability, and impaired DNA repair, which are all characteristics of cancer cells. The nuclear lamina is known to interact with heterochromatins, thereby regulating global genome organization and expression (Dittmer & Misteli, 2011; Van de Vosse et al, 2011). Nuclear dysmorphia is usually accompanied by a loss of nuclear envelope integrity, aberrant gene expression, and micronuclei (Smith et al, 2018). In addition, loss of genomic stability has been regarded as the initiation of tumorigenesis (Sieber et al, 2003). In fact, lamin A has been reported to be involved in the DNA damage response (Gonzalo, 2014). Compared with their wild-type counterparts, mouse embryonic fibroblasts deficient in LMNA show less accumulation of DNA repair factor 53BP1 at DNA damage sites (Redwood et al, 2011). Our results and others together suggest that phosphorylation and disassembly of lamin A by oncogenic Src may be important for it to trigger tumor progression.

More than 400 mutations in *LMNA* have been attributed to at least 11 diseases, which are collectively termed laminopathies (Worman & Bonne, 2007; Kang et al, 2018). The Y45C mutation has been associated with Emery-Dreifuss muscular dystrophy (Bonne et al, 2000). Like the Y45D mutant, the Y45C mutant of lamin A fails to assemble into the nuclear lamina and diffusely distributes in the nucleoplasm (Fig S7). Expressing the Y59C mutant (equivalent to human Y45C) in *Caenorhabditis elegans* inhibited the activation and relocation of muscle-specific promotor (myo-3), which subsequently impaired the differentiation of muscle cells (Mattout et al, 2011). These data suggest that Tyr45 residue is critical for lamin A assembly and muscle-specific gene expression. Substitution of the Tyr45 with certain amino acids may not favor lamin A assembly and possibly lead to pathological consequences. In conclusion, this study not only demonstrated that lamin A is a novel nuclear substrate of Src, but also shed light on the regulation of the nuclear deformability through tyrosine phosphorylation of lamins.

# Materials and Methods

### Materials

The rabbit polyclonal anti-lamin A/C (for immunoprecipitation) was generated using purified His-lamin A (human lamin A a.a.1–646) as the antigen and purified by antigen-affinity column through a custom antibody production service provided by GeneTex, Inc. The rabbit polyclonal antibody specific to lamin A pY45 antibody was generated using synthesized phosphopeptides DRLAVpYIDRVR (Y45 peptide) as the antigen and purified by antigen-affinity double-specific (phosphopeptides and non-phosphopeptides) columns through a custom antibody production service provided by GeneTex, Inc. The mouse monoclonal anti-lamin A (ab8980) that recognizes a.a. 598–611 of lamin A, rabbit polyclonal anti-lamin B1 (ab16375) and anti-lamin B2 (ab155319), and anti-mCherry (ab183628) antibodies were purchased from Abcam. The monoclonal anti-Src antibody (2–17) in mouse ascites generated by a hybridoma (CRL2651) was prepared in our laboratory. The rabbit polyclonal anti-Src pY416 (MAB2685) antibody was purchased from R&D Systems. The mouse monoclonal anti-FLAG (M2) and anti-actin (AC-15) antibodies and

---

***P < 0.001. **(C)** A FRAP analysis was performed in HeLa cells transiently expressing GFP-lamin A WT, Y45F, or Y45D. The selected regions (indicated by yellow arrowhead) were photobleached by a 405-nm laser for 1 s. Representative images from confocal fluorescence microscopy (488 nm excitation) before bleaching (pre-bleach) and at different time points after bleaching are shown. Scale bars, 10 μm. **(D)** Fluorescence at the photobleached regions (indicated by yellow arrowhead) was measured and FRAP was calculated as a ratio to the initial fluorescence. Two-way ANOVAs with Tukey's post hoc tests were used for the comparisons of WT versus Y45D and WT versus Y45F at 20 min time point. The P-values were calculated from at least 18 cells pooled from three independent experiments. ***P < 0.001. **(E)** The changes of fluorescence recovery between 0 and 4 min after photobleaching. The P-values were calculated from at least 18 cells pooled from three independent experiments by t test. *P < 0.05, ***P < 0.001.

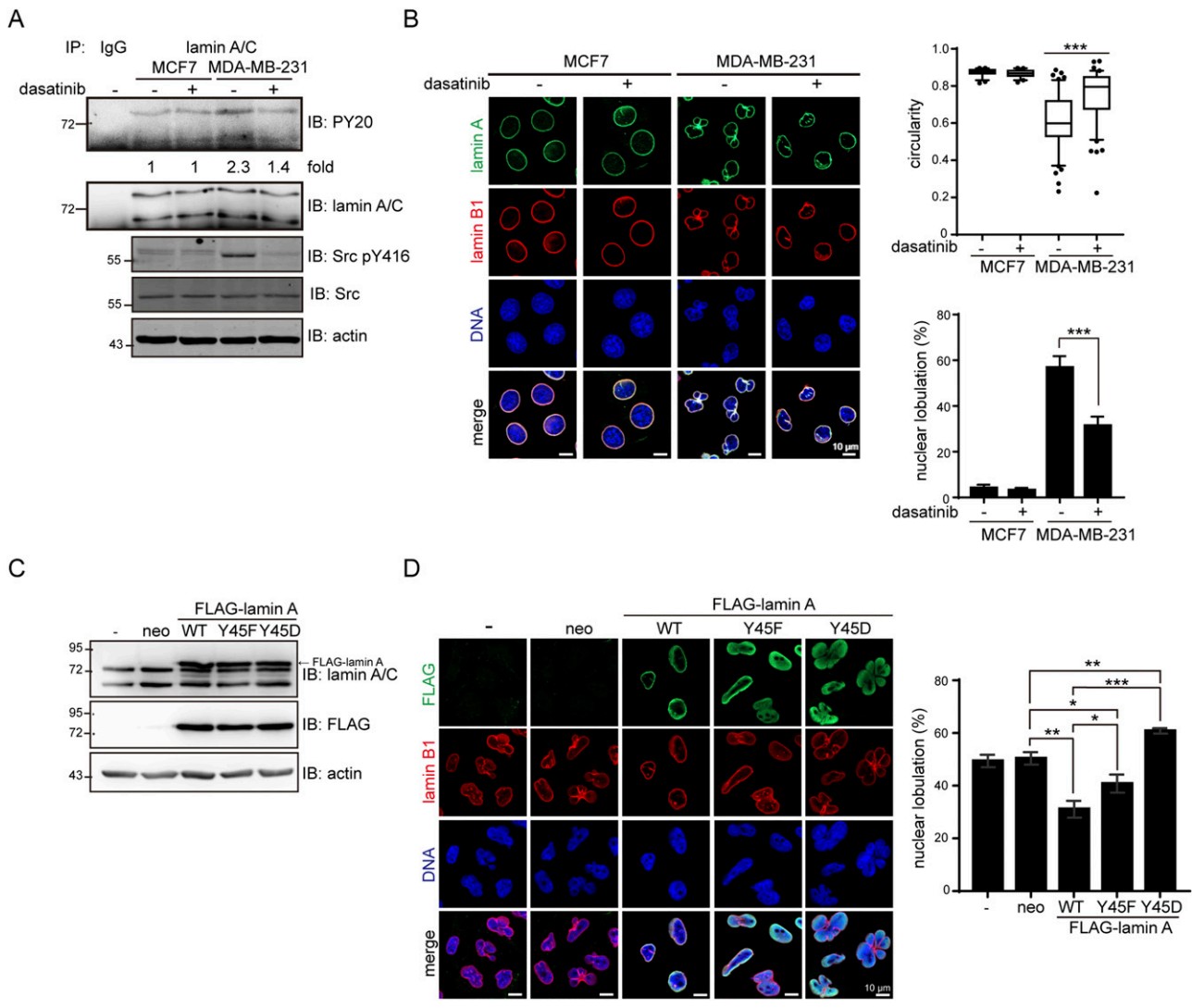

**Figure 5. Aberrant phosphorylation of lamin A by Src may cause nuclear dysmorphia.**
**(A)** MCF7 and MDA-MB-231 cells were treated with (+) or without (−) 50 nM dasatinib for 1 h and then lysed. Equal amounts of the whole cell lysates (WCLs) were incubated with anti-lamin A/C or pre-immune serum (IgG) as the control. The immunonocomplexes were analyzed by immunoblotting (IB) with anti-PY or anti-lamin A/C antibodies. An equal amount of WCLs was analyzed by immunoblotting with anti-Src, anti-Src pY416, or anti-actin. The tyrosine phosphorylation of lamin A was quantified and expressed as -fold relative to the level of MCF7 without dasatinib. **(B)** MCF7 and MDA-MB-231 cells were treated with (+) or without (−) 50 nM dasatinib for 1 h. The cells were fixed and stained for lamin A, lamin B1, and DNA. Representative images are shown. Scale bars, 10 $\mu$m. The nuclear circularity ($4\pi \times$ area/perimeter$^2$) was determined. The P-values were calculated from at least 150 cells pooled from three independent experiments. The percentage of the cells with nuclear lobulation was measured (n ≥ 400). The values (mean ± SD) are from three experiments. ***$P$ < 0.001. **(C)** MDA-MB-231 cells were infected with lentiviruses capable of expressing FLAG-lamin A or the mutants (Y45F and Y45D) and selected in the medium with neomycin (neo). An equal amount of WCLs was analyzed by immunoblotting (IB) with antibodies as indicated. **(D)** The cells as described in panel C were fixed and stained for FLAG-lamin A, lamin B1, and DNA. Representative images are shown. Scale bars, 10 $\mu$m. The percentage of the cells with nuclear lobulation was measured (n ≥ 900). Values (means ± SD) are from three experiments. *$P$ < 0.05, **$P$ < 0.01, ***$P$ < 0.001.
Source data are available for this figure.

protein A–Sepharose beads were purchased from Sigma-Aldrich. The monoclonal anti-phosphotyrosine (PY20) antibody was purchased from BD Transduction Laboratories. The mouse monoclonal anti-GFP (clones 13.1, for immunoprecipitation) antibody was purchased from Roche. The mouse monoclonal anti-γH2AX pS139 (clone JBW301) was purchased from Millipore. The mouse monoclonal anti-GFP (B-2) and rabbit polyclonal anti-emerin (FL254) antibodies were purchased from Santa Cruz Biotechnology. The HRP-conjugated goat anti-mouse and goat anti-rabbit antibodies were purchased from Jackson ImmunoResearch Laboratories. Nocodazole and anti-

FLAG M2 affinity gel were purchased from Sigma-Aldrich. DMEM, Zeocin, Lipofectamine 2000, Alexa Fluor 488–, and Alexa Fluor 546–conjugated secondary antibodies were purchased from Invitrogen Life Technologies. The Src inhibitor dasatinib was purchased from BioVision.

## Plasmids

The plasmids pEVX-cSrc WT and the constitutively active Y527F mutant were described previously (Chan et al, 2003). For GFP-fused

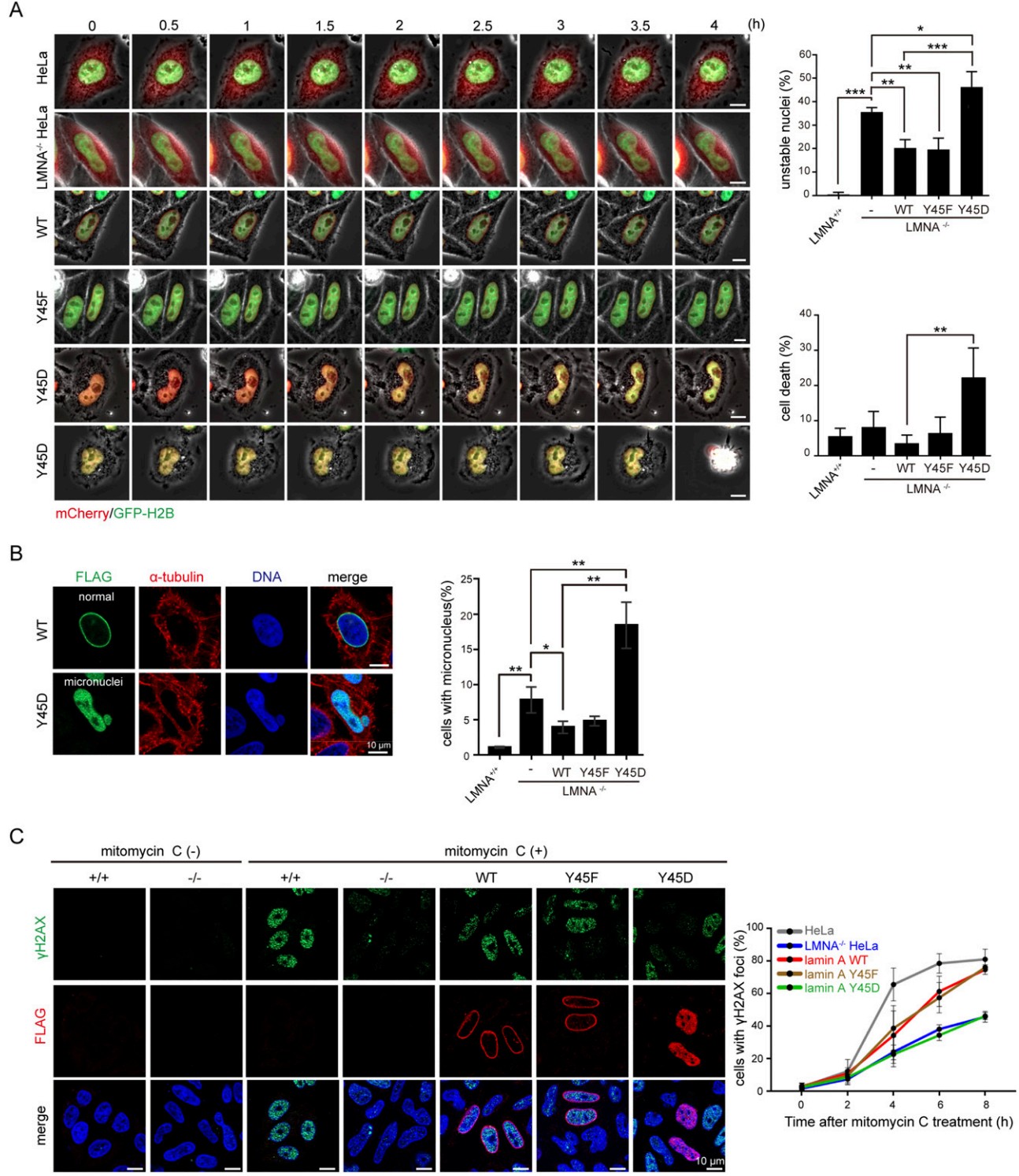

**Figure 6. Aberrant phosphorylation of lamin A by Src may cause genomic instability.**
**(A)** mCherry-lamin A (WT or mutants) and GFP-H2B were transiently co-expressed in LMNA$^{-/-}$ HeLa cells. mCherry vector and GFP-H2B were transiently co-expressed in LMNA$^{+/+}$ and LMNA$^{-/-}$ HeLa cells as the control. The cells were monitored with time-lapse fluorescence microscopy for 24 h. Representative image frames are shown. Scale bars, 10 μm. The percentage of "unstable" nucleus and cell death was measured during the 24 h-period (n ≥ 110). Values (means ± SD) are from three independent experiments. *P < 0.05, **P < 0.01, ***P < 0.001. **(B)** FLAG-lamin A WT or the mutants were transiently expressed in LMNA$^{-/-}$ HeLa cells. The cells were fixed and stained for FLAG, α-tubulin, and DNA. Representative images are shown. Scale bars, 10 μm. The percentage of the cells with micronucleus was determined (n ≥ 500). Values (means ± SD) are from three experiments. *P < 0.05, **P < 0.01. **(C)** FLAG-lamin A WT or the mutants were transiently expressed in LMNA$^{+/+}$ and LMNA$^{-/-}$ HeLa cells. The cells were treated with mitomycin C (1 μM) for different duration as indicated. The cells were fixed and stained for γH2AX pS139, FLAG-lamin A, and DNA. Representative images are shown. Scale bars, 10 μm. For the control LMNA$^{+/+}$ and LMNA$^{-/-}$ HeLa cells, the percentage of the cells with γH2AX pS139 signals were measured (n ≥ 300). For the LMNA$^{-/-}$ HeLa cells transiently expressing FLAG-lamin A or mutants, the percentage of the cells with both FLAG and γH2AX pS139 signals was measured (n ≥ 300). Values (means ± SD) are from three experiments.

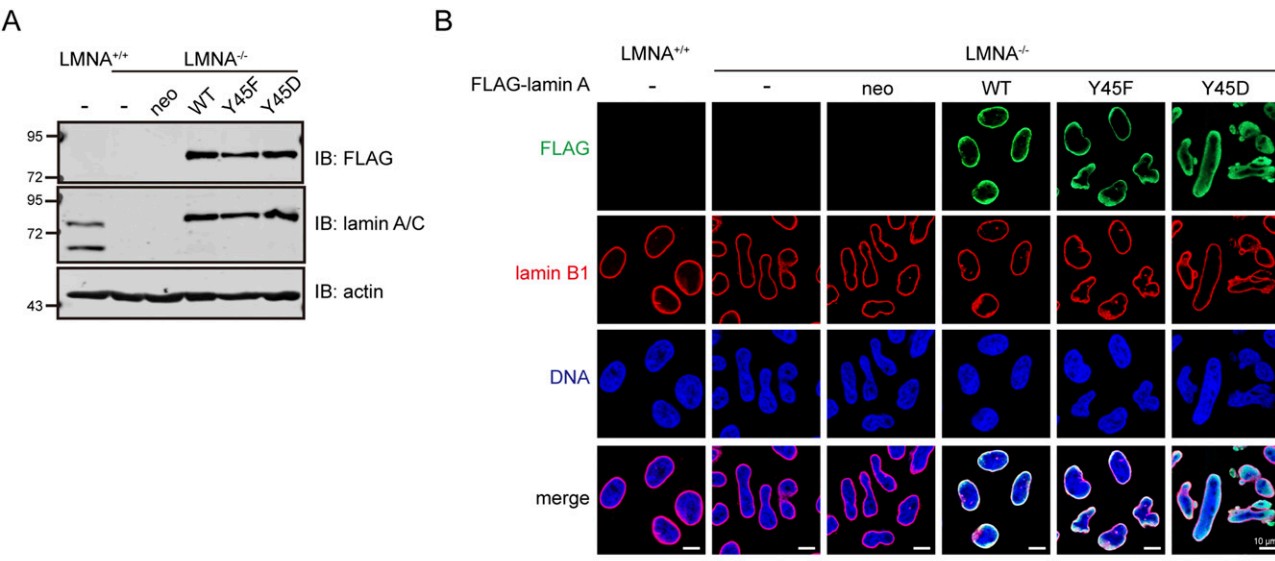

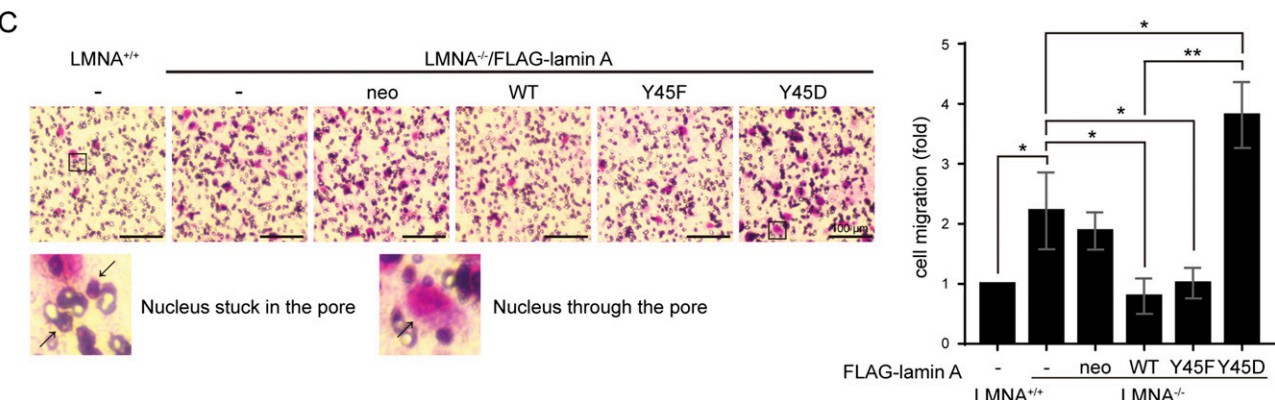

**Figure 7. Aberrant phosphorylation of lamin A at Tyr45 by Src may increase nuclear plasticity for cell migration.**
**(A)** Equal amounts of whole cell lysates from LMNA$^{+/+}$ and LMNA$^{-/-}$ HeLa cells stably expressing FLAG-lamin A were analyzed by immunoblotting (IB) with antibodies as indicated. **(B)** FLAG-lamin A, the mutants or vector alone (neo) were stably expressed in LMNA$^{-/-}$ HeLa cells. The cells were fixed and stained for FLAG-lamin A, lamin B1, and DNA. Representative images are shown. Scale bars, 10 $\mu$m. **(C)** LMNA$^{+/+}$ and LMNA$^{-/-}$ HeLa cells (10$^4$) stably expressing FLAG-lamin A were subjected to a trans-well cell migration assay for 6 h. The cells that migrated to the lower chamber through the membrane with 5-$\mu$m pore size were fixed, stained, and counted. Representative micrographs are shown. Scale bars, 100 $\mu$m. Two enlarge images show the nucleus stuck in the pore or passed through the pore. Data are expressed as –fold relative to LMNA$^{+/+}$ HeLa cells. Values (means ± SD) are from three experiments. *P < 0.05, **P < 0.01.
Source data are available for this figure.

Src, the cDNAs of cSrc, the Y527F mutant, and the kinase-defective (kd) K295R mutant were subcloned into the pEGFP-N3 plasmid using the EcoRI and BamHI sites. For targeting of cSrc to different subcellular localization, the cDNA of the cSrc G2A mutant (defective in the myristoylation) was fused with the NLS (PKKKRKL at N-terminal of Src), the M1 (a.a. 1–66 of the avian infectious bronchitis virus M protein; targeting to the endoplasmic reticulum), or the KDEL receptor (targeting to the Golgi complexes) and then subcloned into pEGFP-N1 plasmid using HindIII and BamHI sites. For HA-cSrc, the cDNAs of cSrc, the Y527F mutant, and the NLS-fused cSrc were subcloned into pcDNA3.1 (+) HA (3) plasmid using BamHI and EcoRI sites. For FLAG-tagged lamin A, the cDNA of human lamin A (a.a.1–646) was cloned into the pCMV-3Tag-3A plasmid using BamHI and XhoI sites. For mCherry-fused lamin A, the cDNA of

human prelamin A (a.a.1–664) was cloned into the mCherry-C1 plasmid using EcoRI and BamHI sites. For GFP-fused lamin A, the cDNA of human prelamin A (a.a. 1–664) was cloned into the pEGFP-C2 plasmid using EcoRI and BamHI sites. For His-tagged lamin A, the cDNA of human lamin A (a.a.1–646) was cloned into the pET-21-b (+) plasmid with the NheI and XhoI sites. All mutagenesis was performed using the QuikChange site-directed mutagenesis kit (Agilent Technologies) and the desired mutations were confirmed by dideoxy DNA sequencing.

### Cell culture and transfection

HEK293, HEK293T, MCF7, MDA-MB-231, and HeLa cells were purchased from the American Type Culture Collection. All the cells were

maintained in DMEM supplemented with 10% fetal bovine serum (Hyclone). For transient transfection, the cells were transfected with 2 µg plasmid DNA through Lipofectamine 2000 and then incubated for 24 h before lysis.

## Generation of lamin A/C-knockout (LMNA$^{-/-}$) HeLa cells

RNA-guided DNA endonuclease was performed to edit genes through co-expression of the Cas9 protein (Addgene plasmid 41815) with gRNAs (http://www.addgene.org/crispr/church/). The targeting sequence (5′-GCGGCGCGCCACCCGCAGCG-3′) for lamin A/C was cloned into the gRNA cloning vector (Addgene plasmid 41824) via the Gibson assembly method (New England Biolabs). Lamin A/C-knockout HeLa cells were obtained through clonal propagation from a single cell. For genotyping, the following PCR primers were used: 5′-CGCACCTACACCAGCCAA-3′ and 5′-CGAACTCACCGCGCTTTC-3′. The PCR products were cloned and sequenced.

## Lentivirus production and infection

For FLAG-lamin A WT, Y45D, and Y45F expression, the respective cDNA was cloned into pLAS3w-pNeo plasmid using NheI and EcoRI sites. For the lentivirus production, HEK293T cells were co-transfected with 0.25 µg pMD.G., 2.25 µg pCMV-ΔR8.91, and 2.5 µg pLAS3w-pNeo-FLAG-lamin A through Lipofectamine 2000. After 3 d, the medium with the viral particles was collected and stored at –80°C. The cells were infected by lentiviruses capable of expressing FLAG-lamin A and selected in the medium containing neomycin (0.5–1.5 µg/ml), respectively. 7 d later, the neomycin-resistant cells were analyzed for FLAG-lamin A by immunoblotting. In this study, stable expression of FLAG-lamin A and its Y45 mutants in LMNA$^{-/-}$ HeLa and MDA-MB-231 cells was established by lentiviral infection.

## Preparation of whole cell lysates

The preparation of whole cell lysates was performed as described previously (Zhang & Sarge, 2008). The cells were suspended in RIPA lysis buffer (1% Nonidet P-40, 50 mM Tris–HCl, pH 7.4, 150 mM NaCl, 1% Na-deoxycholate, 0.1% SDS, 2 mM EDTA, 100 mM NaF, and 1 mM Na$_3$VO$_4$) containing EDTA-free protease inhibitor cocktail (Roche). Cell lysis was performed by sonication with a sonicator (Misonix Sonicator XL2020), after which the sample was incubated on ice for 1 h. After centrifugation at 14,000$g$ at 4°C for 10 min, the supernatant was transferred to a fresh tube and stored at –20°C. Approximately 70% of lamin A was extracted by the method as described above. In Fig 2E, the cell lysates were prepared in 1% NP-40 lysis buffer (1% Nonidet P-40, 20 mM Tris–HCl, pH 7.4, 137 mM NaCl, 10% glycerol, and 1 mM Na$_3$VO$_4$) containing EDTA-free protease inhibitor cocktail (Roche).

## Immunoblotting and immunoprecipitation

Immunoblotting and immunoprecipitation were performed as previously described (Yang et al, 2019). Chemiluminescent detection was performed by a luminescence image system (LAS-4000; Fujifilm).

## Purification of His-tagged lamin A

His-tagged lamin A proteins were expressed in BL21 (DE3) E. coli by induction with 1.0 mM isopropyl β-D-thiogalactopyranoside. The bacterial pellets were lysed in lamin A extraction buffer (8 M urea, 25 mM Tris, pH8.0, and 0.5 M NaCl) with pulsed sonication. The lysates were centrifuged at 14,000$g$ at 4°C for 10 min to remove debris. The supernatants were dialyzed three times with 200 ml of lamin A storage buffer (25 mM Tris, pH 8.0, 0.5 M NaCl, and 1 mM DTT) and stored at –80°C.

## In vitro polymerization of lamin A

Purified His-lamin A (0.5 mg/ml in 100 µl of the storage buffer) was allowed to polymerize by dialysis in the polymerization buffer (25 mM MES, pH 7.0, 150 mM NaCl, and 1 mM DTT) at room temperature for 3 h and followed by centrifugation at 14,000$g$ for 30 min at 4°C. The pellets were dissolved in lamin A polymerization buffer. An equal proportion of His-lamin A in the supernatant and pellet fractions was fractionated by SDS–PAGE and visualized with Coomassie Blue staining. The amount of lamin A polymerization was measured using Image J software.

## In vitro kinase assay

GFP-cSrc Y527F and K295R mutants were transiently expressed in HEK293 cells. The GFP-cSrc was immunoprecipitated by anti-GFP and the immunocomplexes were washed three times with 1% NP-40 lysis buffer and twice with 20 mM Tris buffer, pH 7.4. Kinase reactions were performed in 40 µl of kinase buffer (50 mM Tris–HCl, pH 7.4, and 50 mM MnCl$_2$) containing 100 µM ATP and 0.5 µg purified His-lamin A protein at room temperature for 30 min. The reaction was stopped by the SDS sample buffer, and the proteins were fractionated by SDS–PAGE and analyzed by immunoblotting with anti-phosphotyrosine antibody.

## Immunofluorescence staining

Cells were fixed with 4% paraformaldehyde for 30 min and permeabilized with 0.5% Triton X-100 for 15 min at room temperature. Slides were stained with primary antibodies for 1 h and followed by incubation with Alexa Fluor 488– or 546-conjugated secondary antibodies for 1 h. The primary antibodies, including anti-lamin A (1:500), anti-lamin B1 (1:500), anti-lamin A/C (1:300), anti-HA (1:100), anti-FLAG (1:500), anti-emerin (1: 100), and anti-γH2AX pS139 (1:250) were used in this study. Coverslips were mounted in Dapi-Fluoromount-G (SouthernBiotech). The images were acquired using a Zeiss ApoTome2 microscope imaging system with a Zeiss Plan-Apochromat 40×/NA 1.3, 63×/NA 1.4, or 100×/NA 1.4 oil immersion objective. The images in Fig S3 were acquired using a Zeiss LSM510 microscope imaging system with a Zeiss Plan-Apochromat 63×/ NA 1.4 oil immersion objective. The images were cropped with Photoshop CS6 (Adobe) and assembled into figures with Illustrator CS6 (Adobe).

## FLIP

HeLa cells transiently expressing GFP-lamin A or Y45 mutants were seeded on glass-bottomed dishes. FLIP measurements were

performed under a confocal microscope (LSM 880; Carl Zeiss) in a 37°C and 5% $CO_2$ environment. One scanner was used to bleach a 2-$\mu m$ diameter region for 10 min using a 30-mW 405-nm laser set at 100%, whereas fluorescence images of GFP-lamin A (488-nm laser excitation) were taken at 1 min intervals for 10 min. The loss of fluorescence opposite to the bleached regions was measured. The relative fluorescence ratio in the cytosol of the bleached cells was normalized to the same regions before photobleaching and after background subtraction using ZEISS ZEN2 image software.

### FRAP

HeLa cells transiently expressing GFP-lamin A were seeded on glass-bottomed dishes. FRAP measurements were performed under a confocal microscope (LSM 880; Carl Zeiss) in a 37°C and 5% $CO_2$ environment. A 30-mW 405-nm laser was used to bleach a 1.45 × 5.3 $\mu m$ region at 100% for 1 s, whereas fluorescence images of GFP-lamin A (488-nm laser excitation) were taken at 2 min intervals for 20 min. The recovery of the relative fluorescence ratios was normalized to the same region before photobleaching and analyzed using ZEISS ZEN2 image software.

### Live cell imaging

HeLa or $LMNA^{-/-}$ HeLa cells expressing mCherry vector alone or mCherry-lamin A were incubated in a micro-cultivation system with temperature and $CO_2$ control devices (Carl Zeiss). The cells were monitored on an inverted microscope (Axio Observer; Carl Zeiss) using an EC Plan-NEOFLUAR 40 × NA 0.75 objective. Images were captured every 10 min for 24 h using a digital camera (ORCA-Flash4.0 V2; Hamamatsu) and were processed by ZEISS ZEN2 image software.

### Trans-well migration assay

$LMNA^{+/+}$ and $LMNA^{-/-}$ HeLa cells were collected by trypsinization and suspended in serum-free medium. The experiments were performed in NeuroProbe 48-well chemotaxis chambers. The lower chamber was loaded with serum-free medium with type I collagen (10 $\mu g$/ml). The cells ($10^4$) in serum-free medium were added to the upper chamber. The lower and upper chambers were separated by a polycarbonate membrane (Poretics) with 5 or 8-$\mu m$ pore size. The cells were allowed to migrate for 6 h at 37°C in a humidified atmosphere containing 5% $CO_2$. The membranes were fixed in methanol for 1 h and stained with 10% Giemsa stain for 1 h. The cells that migrated to the lower side of the membrane were counted under a light microscope. Each experiment was performed in triplicate.

### Mass spectrometry

HEK293 cells that transiently co-expressed GFP-lamin A and pEVX-cSrc Y527F were lysed in RIPA lysis buffer containing protease inhibitors. The lysates were centrifuged at 15,000$g$ for 10 min at 4°C. The pellets were solubilized with RIPA lysis buffer and centrifuged again at 15,000$g$ for 10 min at 4°C. The GFP-lamin A in the supernatant was immunoprecipitated with anti-GFP antibody, and the immunocomplexes were fractionated by SDS–PAGE and stained with Coomassie Blue. Mass spectrometry for protein identification and phosphorylation sites was performed as described previously (Yang et al, 2019).

### Densitometric quantitation and statistics

A densitometric quantitation of the scanned images was showed using Image J software (National Institutes of Health). $P$-value was determined by unpaired $t$ tests for two samples and two-way ANOVAs for grouped data. Adobe Illustrator CS6 was used for preparing the figures.

## Supplementary Information

## Acknowledgements

This work was supported by the Ministry of Science and Technology, Taiwan (grant number 107-2923-B010-003-MY3, and 108-2320-B-010-015-MY3) and the Cancer Progression Research Center, National Yang Ming Chiao Tung University from the Featured Areas Research Center Program within the framework of the Higher Education Sprout Project by the Ministry of Education in Taiwan.

### Author Contributions

C-T Chu: data curation, formal analysis, investigation, and writing—original draft.
Y-H Chen: investigation.
W-T Chiu: data curation and investigation.
H-C Chen: conceptualization, supervision, funding acquisition, project administration, and writing—review and editing.

### Conflict of Interest Statement

The authors declare that they have no conflict of interest.

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
