## [Reviewer comments · Life Science Alliance]

Life Science Alliance

Tyrosine phosphorylation of lamin A by Src promotes disassembly of nuclear lamina in interphase

Hong-Chen Chen, Ching-Tung Chu, Yi-Hsuan Chen, and Wen-Tai Chiu

DOI: <https://doi.org/10.26508/lsa.202101120>

Corresponding author(s): Hong-Chen Chen, National Yang Ming Chiao Tung University

Review Timeline:

Submission Date:	2021-05-13
Editorial Decision:	2021-07-08
Revision Received:	2021-07-24
Editorial Decision:	2021-07-28
Revision Received:	2021-08-01
Accepted:	2021-08-03

Scientific Editor: Novella Guidi

Transaction Report:

July 8, 2021

Re: Life Science Alliance manuscript #LSA-2021-01120-T

Author information redacted

Dear Dr. Chen,

Thank you for submitting your manuscript entitled "Tyrosine phosphorylation of lamin A by Src promotes disassembly of nuclear lamina" to Life Science Alliance. The manuscript was assessed by expert reviewers, whose comments are appended to this letter.

As you will note from the reviewers' comments below, both reviewer are quite positive and excited about the work that in their view provides new evidence about the importance of the integrity in the lamins protein at the nuclear structure. They just raised few minor comments that need to be addressed in the manuscript main text and discussion. We, thus, encourage you to submit a revised version of the manuscript back to LSA that responds to all of the reviewers' points including providing some quantification of the phosphorylation of lamin A in figure 5A and increasing the size of the figure 6C.

Thank you for this interesting contribution to Life Science Alliance. We are looking forward to receiving your revised manuscript.

Sincerely,

- A letter addressing the reviewers' comments point by point.
- An editable version of the final text (.DOC or .DOCX) is needed for copyediting (no PDFs).
- High-resolution figure, supplementary figure and video files uploaded as individual files: See our detailed guidelines for preparing your production-ready images, <https://www.life-science-alliance.org/authors>
- Summary blurb (enter in submission system): A short text summarizing in a single sentence the study (max. 200 characters including spaces). This text is used in conjunction with the titles of papers, hence should be informative and complementary to the title and running title. It should describe the context and significance of the findings for a general readership; it should be written in the present tense and refer to the work in the third person. Author names should not be mentioned.

B. MANUSCRIPT ORGANIZATION AND FORMATTING:

Reviewer #1 (Comments to the Authors (Required)):

The submitted manuscript shows that lamin A/C undergoes Tyr45 phosphorylation by Src, which favors its detachment from the peripheral nuclear lamina. The authors claim that Src-mediated phosphorylation favors mitotic solubilization of lamin A/C. Moreover, the authors confirm that lamin A/C absence favors migration in tumor cells. Finally, the authors show the same migratory phenotype in cells expressing phosphorylated lamin A/C (Tyr45). The main finding reported in the submitted paper is that lamin A/C is a Src substrate and that this is relevant to nuclear stability and cell movement.

The paper is well written, with some mistakes in the discussion (please, check all sentences as something is lacking).

Pictures are nice. However, I see that phosphorylated lamin A (Tyr45) is mostly nucleoplasmic, which is a different condition from the mitotic soluble lamin. This must be considered and

extensively discussed by referring to papers from the Roland Foisner group. In my opinion, videos do not add much to the paper.

Reviewer #2 (Comments to the Authors (Required)):

General comments:

In this work the role of the Y45 as one possible mediator in the stability and function of the lamin A protein is demonstrated. The paper shows that modifications on this residue can lead to nuclear dysmorphia, genomic instability and changes in the nuclear plasticity. Overall, the whole work is well founded, and the novel finding is identification of the assembly of the nuclear lamina through the proto-onco gene SRC by the phosphorylation of Y45. This paper represents an impressive amount of work. The number and diversity of the experiments presented here are remarkable. This work provides some exciting new evidence about the importance of the integrity in the lamins protein at the nuclear structure.

Minor comments:

- Paper's Title: Most of the work on this paper is made in proliferating cells, where the expression of the SRC protein is higher during mitosis. I would recommend include this in the paper's title due most of the changes are found in proliferative cells.
- Figure 1C: The authors discard the role SRC-Y527F on lamin B1 using MCF7. Based on the IHC that they are showing I think that changes in lamin b1 structure should not be discarded.
- Figure 1D: in the result section you refer the (WT) SRC as c-SRC, in the legend figure as HA-SRC and the material and method as cSRC this is a little bit confusing for the reader and I recommend use just one nomenclature.
- Figure 2 C: when the author presents the WLC western blot is surprising that they cannot detect any endogenous expression of SRC, when they see some phosphorylation (very light band) in the GFP-lamin A +/- SRC Y527F - lane of the IP.
- Figure 5 A: It looks that dasatinib does not have any effect on the phosphorylation of lamin A in MCF7, how you can explain this? (lane number 3 on the first IB). Also, some quantification of the phosphorylation of lamin A would be appreciated.
- Figure 6 C: Would be recommendable increase the size of this figure. Green channel in the overlay pictures for WT, Y45F and Y45D can barely be seen.
- Discussion: The author mentioned that one possible reason for just lamin A (and not lamin b1 or b2) being phosphorylated is the localization. There is a PNAS paper Nmezi, et al 2019 where they show that the localization of the lamin A is mainly in the inner part of the nuclear lamina. May be this could explain too why SRC can access better to lamin A and not to lamin b that is located more externally.

Reviewer #1

1. The paper is well written, with some mistakes in the discussion

Response:

As suggested by the reviewer, we carefully checked the manuscript, in particular, the discussion section and corrected mistakes.

2. Pictures are nice. However, I see that phosphorylated lamin A (Tyr45) is mostly nucleoplasmic, which is a different condition from the mitotic soluble lamin. This must be considered and extensively discussed by referring to papers from the Roland Foisner group.

Response:

Because the phospho-antibody for Tyr45-phosphorylated lamin A is not suitable for immunofluorescence staining, we did not show any images that represent the nucleoplasmic distribution of Tyr45-phosphorylated lamin A. I am not sure what the reviewer mentioned is based on the distribution of GFP-lamin A Y45D. The phosphomimetic Y45D mutant diffusely distributed in the nucleoplasm (Fig. 3E and 4). However, I concern about whether the distribution pattern of GFP-Y45D mutant can really represent the authentic distribution of endogenous lamin A undergoing Y45 phosphorylation. It is already known that lamin A stochastically redistributes during the cell cycle between the lamina and the nuclear interior, where they tend to localize around nucleoli (Kind et al., 2013; van Steensel and Kind, 2014). In this study, we found that the activity of Src is regulated during the cell cycle, with increased activity in mitosis (Fig. 3J). I think that the role Src-mediated phosphorylation of lamin A in the redistribution between the lamina and the nuclear interior during the cell cycle requires further studies.

3. In my opinion, videos do not add much to the paper.

Response:

I think at least some readers will be of interest in the phenomenon of the “unstable” nucleus induced by depletion of lamin A/C, as shown in Fig. 6A. Therefore, we would like to have those videos as part of supplemented materials.

Reviewer #2

1. Paper's Title: Most of the work on this paper is made in proliferating cells, where the expression of the SRC protein is higher during mitosis. I would recommend include this in the paper's title due most of the changes are found in proliferative cells.

Response:

I agree with the reviewer on this point. Although the notion that phosphorylation of lamin A by Src has an adverse effect on the assembly of nuclear lamina may be also applied in mitosis, we do not have much evidence to support this. Therefore, we would like to change our title as “Tyrosine phosphorylation of lamin A by Src promotes disassembly of nuclear lamina in interphase”.

2. Figure 1C: The authors discard the role SRC-Y527F on lamin B1 using MCF7. Based on the IHC that they are showing I think that changes in lamin b1 structure should not be discarded.

Response:

I agree with the reviewer’s point that the effect of Src-Y527 on lamin B1 should not be discarded. The manuscript relevant to figure 1C was revised.

3. Figure 1D: in the result section you refer the (WT) SRC as c-SRC, in the legend figure as HA-SRC and the material and method as cSRC this is a little bit confusing for the reader and I recommend use just one nomenclature.

Response:

As suggested by the reviewer, we use one nomenclature for HA epitope-tagged c-Src (HA-cSrc) throughout the manuscript.

4. Figure 2 C: when the author presents the WLC western blot is surprising that they cannot detect any endogenous expression of SRC, when they see some phosphorylation (very light band) in the GFP-lamin A +/- SRC Y527F - lane of the IP.

Response:

As suggested by the reviewer, we replaced the blot by a new one showing the expression of endogenous Src.

5. Figure 5 A: It looks that dasatinib does not have any effect on the phosphorylation of lamin A in MCF7, how you can explain this? (lane number 3 on the first IB). Also, some quantification of the phosphorylation of lamin A would be appreciated.

Response:

- (1) The activity of Src in MCF7 cells is relatively low. It is possible that kinase(s) rather than Src may be responsible for the basal tyrosine phosphorylation of lamin A in MCF7 cells, which may explain why dasatinib does not have any effect on the phosphorylation of lamin A in those cells.
- (2) As suggested by the reviewer, we quantified the tyrosine phosphorylation of

lamin A in figure 5A.

6. Figure 6 C: Would be recommendable increase the size of this figure. Green channel in the overlay pictures for WT, Y45F and Y45D can barely be seen.

Response:

As suggested by the reviewer, we increased the size of figure 6C.

7. Discussion: The author mentioned that one possible reason for just lamin A (and not lamin b1 or b2) being phosphorylated is the localization. There is a PNAS paper Nmezi, et al 2019 where they show that the localization of the lamin A is mainly in the inner part of the nuclear lamina. May be this could explain too why SRC can access better to lamin A and not to lamin b that is located more externally.

Response:

As suggested by the reviewer, we add this notion to the discussion.

July 28, 2021

RE: Life Science Alliance Manuscript #LSA-2021-01120-TR

Author information redacted

Dear Dr. Chen,

Thank you for submitting your revised manuscript entitled "Tyrosine phosphorylation of lamin A by Src promotes disassembly of nuclear lamina in interphase". We would be happy to publish your paper in Life Science Alliance pending final revisions necessary to meet our formatting guidelines.

- please upload both main and supplementary figures as single files
- please upload your Tables in editable .doc or excel format
- please add ORCID ID for the corresponding author-you should have received instructions on how to do so
- please add the Twitter handle of your host institute/organization as well as your own or one of the first authors in our system
- please consult our manuscript preparation guidelines <https://www.life-science-alliance.org/manuscript-prep> and make sure your manuscript sections are in the correct order
- please add your main, supplementary figure, and table legends to the main manuscript text after the references section
- please add callouts for Figures S2A, B, and S5A, B to your main manuscript text;

Open questions:

- there is no approval statement and no separate Data availability section, please add if applicable
- panels in Figures 3E, 5D, and S2B are completely black

LSA now encourages authors to provide a 30-60 second video where the study is briefly explained. We will use these videos on social media to promote the published paper and the presenting author. Corresponding or first-authors are welcome to submit the video. Please submit only one video per manuscript. The video can be emailed to contact@life-science-alliance.org

A. FINAL FILES:

B. MANUSCRIPT ORGANIZATION AND FORMATTING:

Sincerely,

August 3, 2021

RE: Life Science Alliance Manuscript #LSA-2021-01120-TRR

Prof. Hong-Chen Chen
National Yang Ming Chiao Tung University
Institute of Biochemistry and Molecular Biology
No. 155, Sec 2, Li-Nong St.
Taipei 11221
Taiwan

Dear Dr. Chen,

Thank you for submitting your Research Article entitled "Tyrosine phosphorylation of lamin A by Src promotes disassembly of nuclear lamina in interphase". It is a pleasure to let you know that your manuscript is now accepted for publication in Life Science Alliance. Congratulations on this interesting work.

DISTRIBUTION OF MATERIALS:

Again, congratulations on a very nice paper. I hope you found the review process to be constructive and are pleased with how the manuscript was handled editorially. We look forward to future exciting submissions from your lab.

Sincerely,
